# Maternal High Linoleic Acid Alters Placental Fatty Acid Composition

**DOI:** 10.3390/nu12082183

**Published:** 2020-07-23

**Authors:** Nirajan Shrestha, Olivia J. Holland, Nykola L. Kent, Anthony V. Perkins, Andrew J. McAinch, James S. M. Cuffe, Deanne H. Hryciw

**Affiliations:** 1School of Medical Science, Griffith University, Southport, QLD 4222, Australia; nirajan.shrestha@griffithuni.edu.au (N.S.); o.holland@griffith.edu.au (O.J.H.); a.perkins@griffith.edu.au (A.V.P.); 2Institute of Health and Biomedical Innovation, Queensland University of Technology, Brisbane, QLD 4000, Australia; 3School of Biomedical Sciences, The University of Queensland, St Lucia, QLD 4067, Australia; nykola.kent@uq.net.au; 4Institute for Health and Sport, Victoria University, Melbourne, VIC 3000, Australia; andrew.mcainch@vu.edu.au; 5Australian Institute for Musculoskeletal Science (AIMSS), Victoria University, St. Albans, VIC 3021, Australia; 6School of Environment and Science, Griffith University, Nathan, QLD 4111, Australia; 7Environmental Futures Research Institute, Griffith University, Nathan, QLD 4111, Australia

**Keywords:** linoleic acid, placenta, maternal diet, fatty acids, nutrient transport

## Abstract

Fetal development is modulated by maternal nutrition during pregnancy. The dietary intake of linoleic acid (LA), an essential dietary n-6 polyunsaturated fatty acid (PUFA), has increased. We previously published that increased LA consumption during pregnancy does not alter offspring or placental weight but fetal plasma fatty acid composition; the developing fetus obtains their required PUFA from the maternal circulation. However, it is unknown if increased maternal linoleic acid alters placental fatty acid storage, metabolism, transport, and general placental function. Female Wistar-Kyoto rats were fed either a low LA diet (LLA; 1.44% of energy from LA) or high LA diet (HLA; 6.21% of energy from LA) for 10 weeks before pregnancy and during gestation. Rats were sacrificed at embryonic day 20 (E20, term = 22 days) and placentae collected. The labyrinth of placentae from one male and one female fetus from each litter were analyzed. High maternal LA consumption increased placental total n-6 and LA concentrations, and decreased total n-3 PUFA, alpha-linolenic acid (ALA), and docosahexaenoic acid (DHA). Fatty acid desaturase 1 (*Fads1*), angiopoietin-like 4 (*Angptl4*), and diacylglycerol lipase beta (*Daglb*) mRNA were downregulated in placentae from offspring from HLA dams. Maternal high LA downregulated the fatty acid transport protein 4 (*Fatp4*) and glucose transporter 1 (*Slc2a1*) mRNA in placentae. IL-7 and IL-10 protein were decreased in placentae from offspring from HLA dams. In conclusion, a high maternal LA diet alters the placental fatty acid composition, inflammatory proteins, and expressions of nutrient transporters, which may program deleterious outcomes in offspring.

## 1. Introduction

Membrane phospholipids contain polyunsaturated fatty acids (PUFA), notably docosahexaenoic acid (DHA) and arachidonic acid (AA) [1]. They act as structural and signaling components of the membrane, and have the capacity to modulate a number of transcription factors, and downstream pathways [1]. In the Western diet, the most abundantly consumed omega (n)-6 PUFA is linoleic acid (LA) [2]. Metabolism of LA can produce AA, which is a potent bioactive molecule that can be converted into eicosanoids by oxidation [3] and involve in endocannabinoid system in the presence of various enzymes such as fatty acid amide hydrolase (*FAAH*), diacylglycerol lipase (*DAGL*)*-α*, *DAGL-β*, and N-acylphosphatidylethanolamines phospholipase D (*NAPE-PLD*) (Figure 1) [4]. Eicosanoids generated by AA oxidation have the capacity to modulate important in physiological processes, such as the immune response, inflammation, fever promotion, and blood pressure regulation/respiration, whereas some are associated with disease development [3]. The n-3 PUFA, alpha-linolenic acid (ALA), is the precursor to DHA, and DHA is highly associated with the nervous system. During development, an increased accumulation of DHA in the infant brain is required for normal neurodevelopment. Metabolism and degradation of LA and ALA occurs via the same enzymes, namely delta-5-desaturase (*Fads1*) and delta-6-desaturase (*Fads2*) [5]. The fetus has insufficient *Fads1* and *Fads2* for the generation of DHA and AA [6], so during pregnancy the fetus obtains PUFA from the maternal diet. In general, PUFA are transported from the maternal to fetal circulation by proteins such as fatty acid transport proteins (FATP) and fatty acid binding proteins (FABP) [7]. More recent research has demonstrated that instead of simple transportation across the placental membrane, fatty acids may be incorporated into placental lipid pools to modulate their transfer to the fetus [8].

We investigated the effects of an elevated concentration of LA during pregnancy on maternal and fetal outcomes. An elevated maternal LA diet during pregnancy altered fatty acid composition and circulated inflammatory proteins in our rodent model. Surprisingly, these changes did not alter fetal weight or placental weight [11]. Of concern for fetal development, a maternal diet with elevated LA altered maternal circulating fatty acids and leptin concentrations [11], and we have evidence that this could lead to altered development [12]. At this time, there has been limited investigations on the capacity of LA to alter placental function. A recent study demonstrated that a high LA diet altered the placental expression of cholesterol packaging and transport genes in the rodent [13]. In this study, there was an elevated content of fat, which may have contributed to the changes observed. Thus, despite the placenta also playing an important role in development [14], at this time we do not know how an elevated maternal LA diet during pregnancy alters fatty acid storage, metabolism and transport. Further, as elevated LA has the capacity to affect proinflammatory protein production which can perturb cellular function, it is important that we understand if an elevated maternal LA diet alters placental function. Developmental programming of disease is often sex-specific, and maternal nutrition is known to affect offspring in a sex-specific manner [15]. Recent research has also demonstrated that the FADS genes are regulated by sex [16]. This suggests that any changes in placental function associated with an elevated maternal LA diet may be sex-specific.

Therefore, the aim of our study was to investigate the sex-specific effects of a high maternal LA diet before and during pregnancy on placental fatty acid storage, metabolism and transport, and general placental function.

## 2. Materials and Methods

### 2.1. Experimental Animal Model and Diet

The animal ethics approved all analyses identified in this study (Griffith University Animal Ethics Committee (NSC/01/17/AEC)). Eight-week-old Wistar-Kyoto (WKY) rats (*n* = 8 for LLA diet and *n* = 9 for HLA diet) were purchased from the Australian Resource Center (ARC, WA, Australia) and housed in accordance to Australian Code of Practice for Care and Use of Animals for Scientific Purpose. All animal experimentation described in the submitted manuscript was conducted in accord with accepted standards of humane animal care, as outlined in the ethical guidelines. 

Rats were housed in individually ventilated cages under 12 h light/12 h dark cycle at a temperature of 20–22 °C and provided with food and water ad libitum throughout the experiment. Female rats were acclimatized for a week and randomly allocated to either a low linoleic acid diet (LLA; 1.44% of energy from LA) or a high linoleic acid diet (HLA; 6.21% of energy from LA) for 10 weeks. These diets were matched for total carbohydrate, total fat, protein, and n-3 FA content. The composition of the diet was detailed in our previous paper [11]. After 8 weeks of dietary exposure, vaginal impedance was monitored daily for at least two estrous cycles using a rat vaginal impedance reader (Muromachi Kikai Co. Ltd., Tokyo, Japan). Female rats were considered ready for mating when vaginal impedance was greater than 4.5 × 10^3^ Ω. Mating occurred following a WKY male rat being placed in the cage with the female rat. Embryonic day 1 (E1) occurred the day following mating. The rats continued to be fed the LLA or HLA diet during pregnancy until embryonic day 20 (E20; term = 22 days). At E20, pregnant female rats were injected with an intraperitoneal (i.p.) injection of sodium pentobarbital (60 mg/kg) form terminal anesthetization. Rats at the time when reflexes were abolished was considered to be indicative of a humane killing. Each fetus was decapitated and had the tail removed and frozen for genotyping of fetal sex. Four placentae from each litter were collected and placentae dissected into the labyrinth and junctional zone, weighed and snap frozen in liquid nitrogen for storage at −80 °C for RNA extraction and lipid extraction (details below). The remaining placenta were fixed whole in 4% paraformaldehyde [17] for histological analysis.

### 2.2. SRY Sex Determination

The fetal sex was determined by genotyping for each fetus [18]. Briefly, the tail was placed in DNA extraction buffer and proteinase K at 55 °C overnight before ammonium acetate was added to each sample. The DNA was precipitated out of solution using isopropanol before it was washed using 70% ethanol and eluted in ultra-pure water. DNA concentration was assessed using the NanoDrop 1000 spectrophotometer (Thermo Fisher Scientific, Seventeen Mile Rocks, QLD, Australia) and 100 ng of DNA was loaded per well into a PCR plate with QuantiNova^TM^ probe master mix (Qiagen, Chadstone, NSW, Australia) and SRY primer probe (Rn04224592_u1; NM_012772.1; Applied Biosystems, Seventeen Mile Rocks, QLD, Australia). TaqMan qPCR was performed using a StepOne^TM^ PCR system (Applied Biosystems, Seventeen Mile Rocks, QLD, Australia).

### 2.3. Fatty Acid Analysis in the Placental Tissue

Placentae from one male and one female per litter were analyzed for fatty acid content. Total lipids for fatty acid analysis were extracted from the labyrinth zone of placental tissue with chloroform/ isopropanol (2:1 *v*/*v*) following the Folch method [19]. Approximately 250 mg of frozen placental tissue was weighed and mixed with 2 mL of 0.9% saline solution. The solution was homogenized and 3 mL of isopropanol was added, vortexed, and allowed to stand for 5 min. Then, 6 mL of chloroform was added, shaken well, and centrifuged for 10 min at 3000× *g*. After this, the bottom organic phase (chloroform) was transferred into a glass scintillation vial which was evaporated to dryness under nitrogen. This was then resuspended in a chloroform: methanol (9:1 *v*/*v*) solution and 20 μL of the sample was spotted onto PUFA-coat^TM^ collection paper [20], allowed to dry at room temperature, and stored for fatty acid analysis. 

The fatty acids in the labyrinth zone of the rat placenta were analyzed using gas chromatography (GC), as previously described [20]. Briefly, samples were transesterified with 2 mL of 1% (*v*/*v*) H_2_SO_4_ in anhydrous methanol and heated for 3 h at 70 °C. The fatty acid methyl esters (FAME) were extracted with heptane and injected into a GC (Hewlett-Packard 6890; Palo Alto, CA, USA) for analysis. FAME were estimated by comparing the retention times and peak area values of unknown samples to those of commercial lipid standards (Nu-check prep Inc., Elysian, MN, USA) using Agilent Chemstation software. The concentration of each fatty acid examined was expressed as a percentage of total lipids.

### 2.4. RNA Extraction, cDNA Synthesis, and Quantitative Polymerase Chain Reaction (qPCR)

RNA from approximately 30 mg of placental labyrinth tissue was extracted from one male and one female per litter using an RNAeasy mini kit (Qiagen, Chadstone, Australia) following manufacturer’s guidelines. Snap frozen placental tissue (between 20 to 30 mg of weight) was homogenized using 600 μL of working RLT buffer mixed with β-mercaptoethanol. Then, the ethanol was added to the homogenized sample and mixed immediately by pipetting. The sample was transferred to a RNeasy spin column placed in a collection tube and the standard step by step centrifugation protocol provided by the manufacturer was executed. RNA concentration was determined using a Nanodrop 1000 spectrophotometer (Thermo Fisher Scientific) and RNA quality was evaluated by 260/280 and 260/230 ratios. cDNA was synthesized from RNA using the iScript gDNA clear cDNA synthesis kit (Bio-Rad Laboratories Inc., Gladesville, NSW, Australia) following manufacturer’s instructions. Quantitative PCR, using QuantiNova SYBR^®^ green master mix (Qiagen), was completed in line with the Minimum Information for Publication of Quantitative Real-Time PCR Experiments (MIQE) guidelines [21]. Heat activation was run for 2 min at 95 °C, then 40 cycles of 95 °C for 5 s, and 60 °C for 10 s using StepOne^TM^ PCR systems (Applied Biosystems). Gene expression was quantified using the 2^−ΔΔCq^ method normalized to the geometric mean of β-actin and β-2 microglobulin as reference genes. These reference genes were stable across the groups. All the primers used for this study were KiCqStart^TM^ predesigned primers from Sigma- Aldrich: β-actin (*Actb*; NM_031144), β-2 microglobulin (*B2m*; NM_012512), fatty acid transport protein 1 (*Fatp1/Slc27a1*; NM_053580.2), fatty acid transport protein 4 (*Fatp4/Slc27a4*; NM_001100706.1), fatty acid binding protein 3 (*Fabp3*; NM_024162.1), fatty acid binding protein 5 (*Fabp5*; NM_145878.1), glutamic-oxaloacetic transaminase 2 (*Got2*; NM_013177.2), *Cd36* (*Fat*, NM_031561.2), lipoprotein lipase (*Lpl*; NM_012598.2), glucose transporter 1 (*Glut1/Slc2a1*; NM_138827.1), glucose transporter 3 (*Glut3/Slc2a3*; NM_017102.2), glucose transporter 4 (*Glut4/Slc2a4*; NM_012751.1), *Slc38a1* (NM_138832.1), *Slc38a2* (NM_181090.2), *Slc38a4* (NM_130748.1), fatty acid desaturase 1 (*Fads1*; NM_053445.2), fatty acid desaturase 2 (*Fads2*; NM_031344.2), vascular endothelial growth factor A (*Vegfa*; NM_031836.3), hypoxia inducible factor 1 subunit alpha (*Hif1a*; NM_024359.1), angiopoietin-like 4 (*Angptl4*; NM_199115.2), fatty acid amide hydrolase (*Faah*; NM_001106678.1), diacylglycerol lipase alpha (*Dagla*; NM_001005886.1), diacylglycerol lipase beta (*Daglb*; NM_001107120.1), and N-acyl phosphatidylethanolamine phospholipase D (*Napepld*; NM_199381.1).

### 2.5. Histological Analysis

Fixed placentae were processed to paraffin and sectioned at a thickness of 7 μM using a microtome. The first three representative midline sections, identified by the presence of the central artery, were collected and stained with hematoxylin and eosin (H-E) [17]. The ScanScope system (Aperio Technologies, Vista, CA, USA) was used to image the sections. General histological appearance was assessed by a blinded observer before this same blinded observer used the Aperio ImageScope (v12.3.0.5056) software to measure the cross-sectional areas of the junctional and labyrinth zones. Total cross-sectional area was determined by adding the average junctional and labyrinth zone cross-sectional areas across the three placental sections from each placenta [17]. 

### 2.6. Multiplex Assay for the Measurement of Cytokines, Chemokines and Growth Factors in Placental Tissue

Placentae from one male and one female per litter were analyzed for the concentrations of cytokines, chemokines, and growth factors in the labyrinth zone. Briefly, protein lysate from placental tissue was extracted using the Bio-Plex^®^ cell lysis kit (Bio-Rad Laboratories, Inc., Gladesville, NSW, Australia). The protein concentration was estimated using a Pierce ^TM^ BCA protein assay kit (ThermoFisher Scientific, Seventeen Mile Rocks, QLD, Australia). The sample was diluted as directed by the manufacturer in Bio-plex sample diluent and a step by step magnetic bead-based multiplex assay was run following the manufacturer’s guidelines. The plate was read using the Bio-Plex 200 system (Bio-Rad Laboratories, Inc., Gladesville, NSW, Australia). All the parameters quantified had an intra-assay coefficient of variation less than 10%. 

### 2.7. Statistical Analysis

All data were analyzed for normality and analyzed separately for male and female outcomes using two-way analysis of variance (ANOVA) with diet and sex as to factors using GraphPad Prism 8.4.0 (GraphPad software, San Diego, CA, USA). One male and one female placenta was used from each litter, with *n* = 8–9. All data are expressed as mean ± standard error of the mean (SEM) and *p* < 0.05 was considered statistically significant. 

## 3. Results

### 3.1. Effect of High Maternal LA Diet on Fatty Acid Composition in Rat Placenta

Composition of fatty acids is expressed as percentage of total lipid plasma fatty acids. High maternal LA diet significantly increased total SFA in placental labyrinth zone (effect of diet *p* < 0.0001) (Table 1). Total monounsaturated fatty acid (MUFA) was decreased in placental labyrinth from the offspring of dams fed with HLA diet (effect of diet *p* < 0.0001). Total n-9 FA, total n-7 FA, total n-3 FA, ALA (18:3n-3), and docosahexaenoic acid (DHA; 22:6n-3) were significantly decreased in placental labyrinth from the offspring of dams fed with HLA diet compared to that fed with LLA diet (effect of diet *p* < 0.0001). As expected, a high maternal LA diet significantly elevated total n-6 FA and LA (18:2n-6) in placental labyrinth (effect of diet *p* < 0.0001). Surprisingly, AA, the biochemical metabolite of LA remained unchanged among the groups in labyrinth zone of placenta, with no sex differences in the fatty acid composition.

### 3.2. Effect of High Maternal LA Diet on Gene Expressions of Enzymes Involved in Fatty Acid Metabolism and Transport in Labyrinth Zone of Rat Placenta

*Fads1* and *Fads2* are major enzymes involved in the desaturation of LA and ALA to produce longer chain PUFAs. The mRNA expression of *Fads1* was significantly downregulated in the placental labyrinth of offspring from dams consuming a HLA diet (effect of diet *p* = 0.01) (Table 2). *Fads2* remained unchanged among the groups. We measured the relative mRNA expression of enzymes involved in the fatty acid metabolism in the labyrinth zone of placenta. There were no differences in the mRNA expressions of *Faah*, *Dagla*, and *Napepld* (Table 2). Maternal HLA diet significantly decreased the mRNA expression of *Daglb* in the labyrinth zone of placenta (effect of diet *p* = 0.04). Maternal nutrition has a major impact on placental nutrient transporters; hence, we next investigated the effect of high maternal LA diet on gene expression of fatty acid transporters in labyrinth zone of the placenta (Table 2). The mRNA expression of *Fatp4* was significantly downregulated (effect of diet *p* = 0.03) in placental labyrinth of offspring from dams fed with HLA. There were no changes in the expression of other genes related to fatty acid transport in rat placenta such as *Fatp1*, *Fabp3*, *Fabp5*, *Got2*, *Cd36*, and *Lpl* among the groups. There were no sex-specific changes in the expression of genes associated with fatty acid transport in the rat placenta.

### 3.3. Effect of High Maternal LA Diet on Placental Morphology

Despite no change in overall placental weight [11], LA may still have the capacity to negatively affect placental morphology. H-E staining demonstrated no effect of HLA on gross morphology, with no overt changes to specific cell types, blood space areas, or increased rate of overt abnormalities. Furthermore, total cross-sectional area of the placentas was not affected by treatment (Figure 2). Analysis of the placental zone cross-sectional areas (labyrinth and junctional zones) relative to the whole placenta cross sectional area demonstrated no changes between treatments in the size of zones.

### 3.4. Effect of High Maternal LA Diet on the Expressions of Genes Related to Angiogenesis, Glucose Transport, and Amino Acid Transport in Labyrinth Zone of Rat Placenta

Placental angiogenesis is critical for this organ’s function [22] and no change in weight or structure may not be indicative of placental angiogenesis. To investigate the effect of a maternal HLA diet on placental angiogenesis, we evaluated the mRNA expression of genes involved in angiogenesis; namely *Vegfa*, *Hif1a*, and *Angptl4* in the labyrinth zone of rat placenta (Table 3). A high maternal LA diet did not affect the mRNA expression of *Vegfa* and *Hif1a* in the placental labyrinth, yet HLA significantly decreased the mRNA expression of *Angptl4* (effect of diet *p* = 0.014) in the labyrinth zone of placenta. Because fatty acids alter glucose and amino transport [23,24], we measured the expression of genes related to glucose and amino acid transport. The mRNA expression of glucose transporter 1 (*Slc2a1*) was lower in placental labyrinth of offspring from dams consuming a HLA diet compared to offspring from dams fed a LLA diet (effect of diet *p* = 0.0015). The mRNA expressions of other genes related to glucose transport namely *Slc2a3* and *Slc2a4* were not different between the groups. Further, there were no changes in the gene expression of *Slc38a1*, *Slc38a2*, and *Slc38a4*, which are involved in amino acid transport and no changes were sex-specific.

### 3.5. Effect of High Maternal LA Diet on Cytokines, Chemokine, and Growth Factors in Labyrinth Zone of Rat Placenta

Fatty acids have the potential to modulate inflammatory mediators [25] and we previously demonstrated that hepatic maternal inflammatory mediators were altered in response to maternal LA consumption, despite no changes in circulating maternal inflammatory mediators [11]. In this study, we show that there were no changes in the concentrations of interleukins (IL-1β, IL-2, IL-5, IL-6), interferon-gamma (IFN-γ), tumor necrosis factor-alpha (TNF-α), vascular endothelial growth factor (VEGF), and monocyte chemoattractant protein 1 (MCP-1) among the groups in the placental labyrinth zone (Table 4). Diet and sex have an interactive effect on the level of IL-1α in placental labyrinth zone. The concentrations of IL-7 and IL-10 significantly decreased in placental labyrinth of offspring from dams fed with HLA diet. Maternal diet and sex have interactive effect on the concentration of regulated on activation, normal T cell express and secreted (RANTES) in placental labyrinth zone (effect of interaction of diet and sex *p* = 0.008).

## 4. Discussion

Maternal nutrition has a major role in gestation and the long term health of the offspring through fetal programing which are often mediated by placental adaptations [26]. In this study, we demonstrated that a high maternal LA diet alters the placental fatty acid composition, genes responsible for the metabolism, transport of fatty acids, genes responsible for nutrient transport, and angiogenic factors. There is a clear difference in placental fatty acid composition in the placenta from offspring exposed to a high maternal LA diet. Surprisingly, there are few differences in the expression of genes related to nutrient transport and angiogenesis, and no sex-specific differences observed. Notably, this is the first study to investigate the effect of an elevated maternal LA diet on placental fatty acid composition and gene expression related to nutrient transport and angiogenesis, using a diet which is similar to the Australian diet. Previous studies have investigated the effects of an elevated LA diet using a significantly higher LA concentration. Further, a high maternal LA diet alters inflammatory mediators in a sex-specific manner. Collectively, these may program deleterious outcomes in offspring.

Pregnancy related complications are associated with alterations of fatty acid composition in the maternal circulation [27]. Inadequate maternal PUFA dietary intake is associated with altered placental fatty composition and expression of FATPs [28]. A recent study reported that total fat content and LA/ALA ratio in the maternal diet influenced the placental fatty acid composition in rat model [29]. Similarly, we previously demonstrated that a high maternal LA diet alters maternal and fetal plasma fatty acid composition [11]. Of note, the pattern of change in placental fatty acid composition was similar to both the maternal and fetal circulations, with some exceptions. For example, a high maternal LA diet increased the level of total SFA in both male and female placentae, despite the matched SFA in the maternal diet. There were similar patterns of change in the fetal plasma, however, there was no changes in maternal plasma level of SFA in our previous study [11]. Interestingly, in the present study, total n-3 PUFA, ALA, and DHA were decreased in the placentae of offspring from dams fed with HLA diet. It is important to note that this difference in fatty acid content in the placentae occurred despite matched total n-3 PUFA in the experimental diet of both groups. This is consistent with a previous finding that the dams exposed to high LA diets had lower placental DHA compared to those fed with low LA diet in the rat [29]. These findings suggest altered metabolism of LA and ALA by the placenta in response to an elevated maternal LA diet.

As n-6 FA and LA compositions were increased in the placentae of offspring from dams fed with the HLA diet, we hypothesized that placental AA should also be increased. In our previous study, we observed that a high maternal LA diet elevates the AA concentrations both in maternal and fetal plasma [11]. However, in the current study (where we used the placentae from the same animal model [11]), the proportion of AA in the placenta remained unchanged between the groups. AA is a longer chain n-6 PUFA which is generated from LA, through *FADS1* and *FADS2* activity [2]. AA is the most important PUFA associated with membrane phospholipids. Upon release, AA can be enzymatically metabolized into eicosanoids, which is associated with disease, in addition to tissue homeostasis [3]. Previous research has demonstrated that AA is rapidly transfer from the maternal to fetal circulation [30], however the elevated maternal and fetal concentrations, but stable placental concentrations leads to the proposal that the placental is reducing the effect of AA on the placental cells, but modulating the AA concentration, irrespective of its source. 

LA and ALA are converted to longer chain PUFA by a consecutive series of reactions, including desaturation in the presence of *Fads1* and *Fads2*, and elongation and oxidation [16]. Maternal and fetal blood fatty acid compositions are substantially influenced by the Fads1 genotype in Japanese population [31]. Further, there is also an association between n-6 PUFA and *Fads2* polymorphisms in pregnant women [32]. Specifically, that both maternal and child FADS genotypes and haplotypes have the capacity to influence the cord plasma PUFA concentrations and fatty acid ratios [32]. In the present study, the mRNA expression of *Fads1* was downregulated in the placentae of offspring from dams fed with HLA diet. The decreased level of DHA, a longer chain n-3 PUFA, in the placentae in the current study may be associated with downregulation of the expression of *Fads1*. Given that LA can be metabolized to form endocannabinoids [2], we also investigated the role of high maternal LA diet on the placental endocannabinoid system (ECS). The two main endocannabinoids, anandamide (AEA) and 2-arachidonyl glycerol (2-AG) are derivatives of dietary LA [2]. These endocannabinoids are synthesized and degraded in the presence of enzymes namely- FAAH, DAGL-α, DAGL-β, and NAPE-PLD. Studies have shown that these endocannabinoids are important mediators of placentation [33]. Recently, we have reported that high maternal LA alters the ECS in maternal and fetal cardiac tissue [12]. The gene expression of *Daglb* was decreased in the placentae from offspring from dams fed a HLA diet. However, we found no effect on the gene expressions of other enzymes involved in ECS in placenta (*Faah*, *Dagla*, *Napepld*). 

At this time, we do not have a clear understanding about which fatty acid transporters bind and transport fatty acids in the placenta. In the present study, a high maternal LA diet decreased the gene expression of *Fatp4* in the placentae exposed to an elevated maternal LA diet. A previous study reported that an increased n-6/n-3 ratio in the diet increased the porcine gene expression of *Fatp1* in the longissimus dorsi muscle and subcutaneous adipose tissue [34], yet this is the first study to report the role of a maternal high LA diet on placental gene expression of FATPs and FABPs. In a rat model, *Fatp1*, *Cd36*, and *Fabp3* mRNA were decreased in a high fat diet (HFD) exposed placentae [35]. Of note, fat content was matched in our experimental model and there was no changes in maternal body weight, as reported in our previous study [11]. 

In our previous study, we demonstrated that a high maternal LA diet altered fetal circulating fatty acid concentration, in the absence of changes to fetal and placental weights [11]. Weight changes may not be indicative of functional changes. The placental labyrinth zone is key for maternal/fetal exchange and a key finding of this study is that a high maternal LA diet alters expression of nutrient transporters in placental labyrinth at gestation day 20. Recent studies have investigated the effect of different concentrations of LA in the maternal diet on placental weight, fatty acid composition and expression of genes associated to cholesterol transport and packaging [13,29]. However, to the best of our knowledge, this is the first study to investigate the effect of a high maternal LA diet on nutrient transport and angiogenesis associated genes.

Increased plasma free fatty acid concentrations reduce glucose uptake mediated by insulin, which may be mediated through a reduction in glucose transport [36]. We hypothesized that if a high maternal LA diet increased the fatty acid composition within the maternal blood, this would result in altered placental expression of genes related to glucose transport. We observed that a high maternal LA diet decreased the expression of glucose transporter 1 (*GLUT1*; *Slc2a1*) in the placentae. A previous human study has demonstrated that maternal obesity has the capacity to alter the expression of placental glucose transporters *Slc2a1* and circulating offspring glucose concentrations [37]. However, in our current rat model, elevated maternal LA does not change fetal glucose concentrations [11], but as outlined here does alter *Slc2a1* expression in placenta. This does not negate a potential effect on glucose released in response to a challenge in these offspring. Previous studies have also reported that a range of maternal perturbations impair placental amino acid transporter expression. Maternal high fat feeding has been shown to upregulate the expression of system A amino acid transporters in the placenta [38]. However, a high maternal LA diet seems to have no effect on genes related to amino acid transport in placentae.

Placental angiogenesis is critical for normal fetal growth and development [39]. Diet induced obesity has been shown to alter placental function and inhibit placental angiogenesis in a mouse model [40]. Angiogenic factors include the vascular endothelial growth factor (VEGF), fibroblast growth factor (FGF), angiopoietin (ANG) protein families, and hypoxia-inducible factor 1 (HIF-1), as well as their respective receptors [39,41]. In the current study, we found no overt morphological changes suggestive of altered vasculogenesis. Similarly, HLA had no effect on the expression of *Vegfa* and *Hif1a*, yet the mRNA expression of *Angptl4* significantly decreased in placentae of offspring from dams fed with an HLA diet. A previous study that utilized placental derived mesenchymal stromal cells, demonstrated that treatment of n-3 PUFA upregulate the expression of VEGF suggesting that n-3 PUFA may promote angiogenesis in the placenta [42]. In contrast, a number of studies have reported the anti-angiogenic properties of n-3 PUFA [43,44]. Studies which have investigated n-6 PUFA have reported stimulatory or neutral effects on angiogenesis [44,45]. In addition, cis-9, trans-11 conjugated LA stimulates the expression of *Angptl4* in placental extravillous trophoblast cells [44]. In the current study, high maternal LA had only minor effects on angiogenic factors in the placenta with inhibition of *Angptl4* expression in the placentae the only change identified. 

We have previously shown that a high maternal LA diet alters maternal hepatic inflammatory cytokines [11]. However, few studies have specifically investigated the role of maternal PUFA consumption on placental inflammation, with those studies that have been performed largely focused on n-3 PUFA intake [46,47]. In the current study, a high maternal LA diet decreased the concentrations of IL-7 and IL-10 in placentae from offspring from HLA dams. Thus, a high maternal LA diet seems to have both pro-inflammatory and anti-inflammatory properties in the placenta, as IL-7 possesses pro-inflammatory properties [48] and IL-10 is an anti-inflammatory cytokine [49]. Further, the sex of the offspring determines the effect of the HLA diet on the concentration of Il-1 and RANTES in placenta. RANTES can be downregulated by n-3 [50], yet despite both placentae from males and females born to dams consuming high LA, the male placenta shows reduced concentration of RANTES, while the female has elevated concentration. This suggests that expression of some inflammatory pathways may be sex-specific in the placentae from offspring from HLA dams. Similar to animal studies, randomized control trials and systematic review, it appears that LA has a dual role in inflammation [51,52].

In conclusion, the current study demonstrates the impact of a high maternal LA diet on placental fatty acid composition and genes responsible for metabolism and transport, in addition to genes involved in nutrient transport and angiogenesis. Further, we show that a high maternal LA diet influenced the placental inflammatory responses. These changes suggest that the placenta exposed to a high maternal LA diet has altered handling of fatty acids and potentially nutrient transport. As the in-utero environment is critical for development, the present study emphasizes the importance of exploring the role of high maternal LA diet on health in adult offspring.

## Figures and Tables

**Figure 1 nutrients-12-02183-f001:**
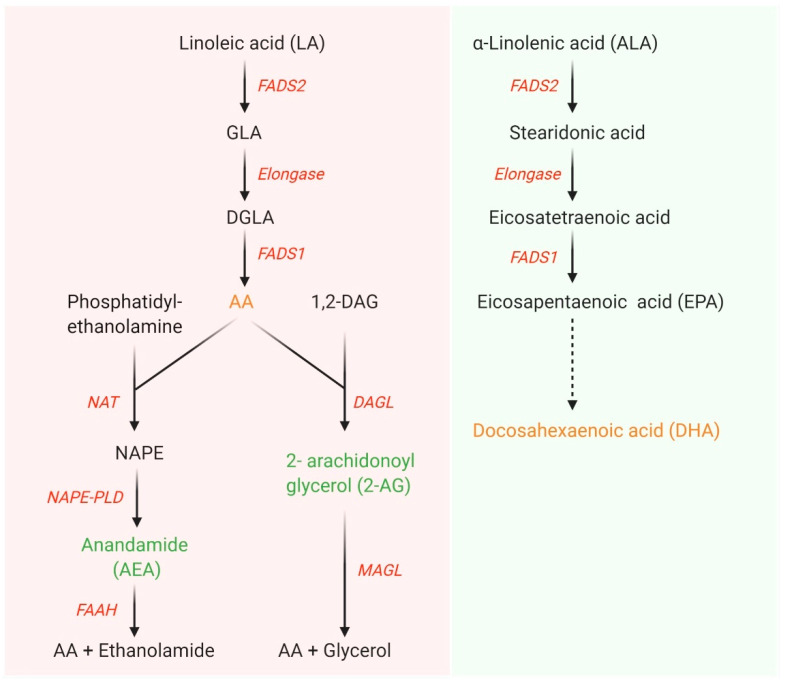
Metabolism of linoleic acid (LA) and α-linolenic acid (ALA). LA and ALA are converted into longer chain polyunsaturated fatty acids in the presence of series of enzymes namely Fads1, Fads2, and elongase. Arachidonic acid (AA) produced from LA involves in the endocannabinoid system pathway. DGLA, dihomo-gamma-linolenic acid; GLA, gamma-linolenic acid; FADS, fatty acid desaturase; 1,2-DAG, 1,2-diacylglycerol; NAPE, N-arachidonoylphosphatidylethanolamine; NAPE-PLD, NAPE-phospholipase D; FAAH, fatty acid amide hydrolase; NAT, N-acyltransferase; DAGL, diacylglycerol lipase; MAGL, monoacylglycerol lipase. Figure adapted from [9,10] and prepared in Biorender.

**Figure 2 nutrients-12-02183-f002:**
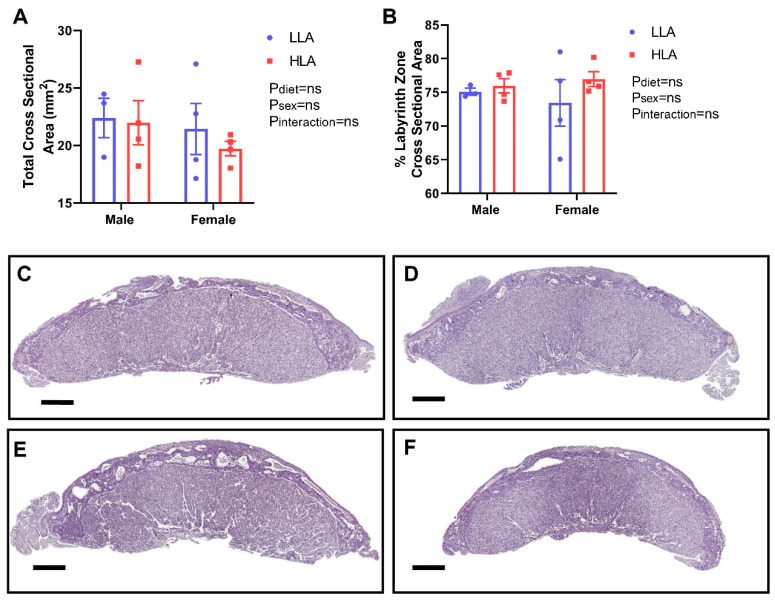
The effect of maternal diet high in linoleic acid on gross placental morphology at embryonic day 20. (**A**) Average total cross-sectional area (mm^2^) in male and female placentae. (**B**) Percentage labyrinth zone of total cross-sectional area in male and female placentae. (**C**) Representative placental sections of LLA male placenta. (**D**) Representative placental sections of LLA female placenta (**E**) Representative placental sections of HLA male placenta. (**F**) Representative placental sections of HLA female placenta stained with H-E at 1× magnification (scale bar = 1000 µm). *n* = 3–4. Data are presented as mean ± SEM with each dot on the scatter plot symbolizing a single representative sample taken from each litter. Data was analyzed using a two-way ANOVA with diet and sex as two factors. LLA = low linoleic acid; HLA = high linoleic acid.

**Table 1 nutrients-12-02183-t001:** Effect of a maternal diet high in linoleic acid on fatty acid composition (% of total lipid plasma fatty acid) in the labyrinth zone of the rat placenta.

	LLA		HLA		*p*-Values		
	Male	Female	Male	Female	Diet	Sex	Interaction
Total SFA	50.98 ± 1.16	52.43 ± 1.27	57.42 ± 0.97	56.04 ± 0.51	*p* < 0.0001	ns	ns
Total MUFA	17.56 ± 0.38	16.88 ± 0.37	8.97 ± 0.29	9.17 ± 0.22	*p* < 0.0001	ns	ns
Total n-9 FA	14.43 ± 0.38	13.79 ± 0.37	6.71 ± 0.24	6.9 ± 0.17	*p* < 0.0001	ns	ns
Total n-7 FA	3.13 ± 0.02	3.06 ± 0.03	2.24 ± 0.10	2.28 ± 0.05	*p* < 0.0001	ns	ns
Total n-3 FA	2.91 ± 0.08	3.05 ± 0.20	1.92 ± 0.12	1.77 ± 0.09	*p* < 0.0001	ns	ns
18:3n-3 (ALA)	0.3 ± 0.07	0.18 ± 0.06	0.00 ± 0.0	0.00 ± 0.0	*p* < 0.0001	ns	ns
22:6n-3 (DHA)	2.51 ± 0.12	2.7 ± 0.13	1.92 ± 0.12	1.77 ± 0.09	*p* < 0.0001	ns	ns
Total n-6 FA	28.35 ± 0.93	27.38 ± 0.83	31.57 ± 0.99	32.93 ± 0.51	*p* < 0.0001	ns	ns
18:2n-6 (LA)	9.55 ± 0.40	9.17 ± 0.27	12.62 ± 0.63	13.1 ± 0.62	*p* < 0.0001	ns	ns
20:4n-6 (AA)	16.39 ± 0.54	16.19 ± 0.37	15.91 ± 0.34	16.36 ± 0.20	ns	ns	ns

LLA: low linoleic acid; HLA: high linoleic acid; SFA: saturated fatty acid; MUFA: monounsaturated fatty acid; LA: linoleic acid; AA: arachidonic acid. Placentae from one male and one female per litter were analyzed. Data expressed as mean ± standard error of the mean (SEM) and analyzed by two-way ANOVA with diet and sex as two factors. *n* = 8 (LLA) and *n* = 9 (HLA).

**Table 2 nutrients-12-02183-t002:** Effect of a maternal diet high in linoleic acid on relative mRNA expression of genes related to fatty acid metabolism and transport in the labyrinth zone of the rat placenta.

	LLA		HLA		*p*-Values		
	Male	Female	Male	Female	Diet	Sex	Interaction
**Genes associated with fatty acid metabolism**							
*Fads1*	1.04 ± 0.11	1.07 ± 0.19	0.63 ± 0.06	0.83 ± 0.10	*p* = 0.010	ns	ns
*Fads2*	1.08 ± 0.18	1.10 ± 0.22	1.08 ± 0.12	1.30 ± 0.11	ns	ns	ns
*Faah*	1.13 ± 0.20	1.14 ± 0.28	0.60 ± 0.14	0.85 ± 0.19	ns	ns	ns
*Dagla*	1.22 ± 0.37	1.09 ± 0.22	0.78 ± 0.22	0.75 ± 0.13	ns	ns	ns
*Daglb*	1.08 ± 0.17	1.06 ± 0.17	0.63 ± 0.15	0.83 ± 0.14	*p* = 0.04	ns	ns
*Napepld*	1.08 ± 0.15	1.08 ± 0.21	0.96 ± 0.09	0.98 ± 0.06	ns	ns	ns
**Genes associated with fatty acid transport**							
*Fatp1*	1.07 ± 0.14	1.06 ± 0.16	0.61 ± 0.13	1.03 ± 0.21	ns	ns	ns
*Fatp4*	0.86 ± 0.07	1.23 ± 0.42	0.59 ± 0.10	0.67 ± 0.09	*p* = 0.03	ns	ns
*Fabp3*	1.02 ± 0.08	1.04 ± 0.13	1.32 ± 0.12	1.11 ± 0.10	ns	ns	ns
*Fabp5*	1.02 ± 0.09	1.04 ± 0.14	1.16 ± 0.07	1.07 ± 0.09	ns	ns	ns
*Got2*	1.12 ± 0.21	1.14 ± 0.24	0.96 ± 0.19	0.95 ± 0.14	ns	ns	ns
*Cd36*	1.05 ± 0.13	0.85 ± 0.05	0.86 ± 0.05	0.90 ± 0.07	ns	ns	ns
*Lpl*	1.06 ± 0.13	1.04 ± 0.15	0.85 ± 0.06	0.97 ± 0.06	ns	ns	ns

LLA: low linoleic acid; HLA: high linoleic acid; FADS1: fatty acid desaturase 1 (also known as delta-5 desaturase); FADS2: fatty acid desaturase 2 (also known as delta-6-desaturase FAAH: fatty acid amide hydrolase; DAGLA: diacylglycerol lipase alpha; DAGLB: diacylglycerol lipase beta; NAPEPLD: N-acyl phosphatidylethanolamine phospholipase D. FATP: fatty acid transport protein; FABP: fatty acid binding protein; GOT2: glutamic-oxaloacetic transaminase 2; CD36: cluster of differentiation 36 (also known as fatty acid translocase); LPL: lipoprotein lipase. Placentae from one male and one female per litter were analyzed Data expressed as mean ± SEM and analyzed by two-way ANOVA with diet and sex as two factors. *n* = 5–8.

**Table 3 nutrients-12-02183-t003:** Effect of a maternal diet high in linoleic acid on relative mRNA expression of genes related to angiogenesis, glucose transport and amino acid transport in the labyrinth zone of the rat placenta.

	LLA		HLA		*p*-Values		
	Male	Female	Male	Female	Diet	Sex	Interaction
**Genes related to angiogenesis**							
*Vegfa*	1.05 ± 0.13	1.13 ± 0.29	0.95 ± 0.07	0.96 ± 0.07	ns	ns	ns
*Hif1a*	1.30 ± 0.34	1.32 ± 0.31	0.92 ± 0.15	1.50 ± 0.22	ns	ns	ns
*Angptl4*	0.89 ± 0.06	1.11 ± 0.23	0.49 ± 0.11	0.71 ± 0.16	*p* = 0.014	ns	ns
**Genes related to glucose transport**							
*Slc2a1*	1.02 ± 0.08	1.05 ± 0.17	0.61 ± 0.11	0.64 ± 0.09	*p* = 0.0015	ns	ns
*Slc2a3*	1.10 ± 0.18	1.06 ± 0.16	0.80 ± 0.13	0.88 ± 0.12	ns	ns	ns
*Slc2a4*	1.11 ± 0.19	1.07 ± 0.17	0.70 ± 0.20	0.79 ± 0.18	ns	ns	ns
**Genes related to amino acid transport**							
*Slc38a1*	1.05 ± 0.13	1.02 ± 0.10	0.81 ± 0.11	0.95 ± 0.09	ns	ns	ns
*Slc38a2*	1.10 ± 0.17	1.11 ± 0.25	0.78 ± 0.08	0.93 ± 0.12	ns	ns	ns
*Slc38a4*	1.03 ± 0.10	1.02 ± 0.11	0.88 ± 0.10	0.93 ± 0.07	ns	ns	ns

LLA: low linoleic acid; HLA: high linoleic acid; VEGFA: vascular endothelial growth factor A; HIF1A: hypoxia inducible factor 1 subunit alpha; ANGPTL4: angiopoietin like 4. Placentae from one male and one female per litter were analyzed. Data expressed as mean ± SEM and analyzed by two-way ANOVA with diet and sex as two factors. *n* = 5–8.

**Table 4 nutrients-12-02183-t004:** Effect of a maternal diet high in linoleic acid on cytokines, chemokines, and growth factors in the labyrinth zone of the rat placenta.

	LLA		HLA		*p*-Values		
	Male	Female	Male	Female	Diet	Sex	Interaction
IL-1α (pg/mL)	10.6 ± 1.56	8.0 ± 1.22	8.0 ± 0.75	11.1 ± 1.17	ns	ns	*p* = 0.03
IL-1β (pg/mL)	45.4 ± 6.38	85.5 ± 30.01	50.3 ± 4.86	45.5 ± 1.93	ns	ns	ns
IL-2 (pg/mL)	987.1 ± 103.7	1411.0 ± 303.3	935.4 ± 65.2	972.5 ± 155.9	ns	ns	ns
IL-5 (pg/mL)	57.5 ± 7.27	57.0 ± 12.58	39.73 ± 4.40	56.4 ± 8.73	ns	ns	ns
IL-6 (pg/mL)	54.8 ± 7.30	38.9 ± 7.72	34.7 ± 5.34	37.8 ± 6.64	ns	ns	ns
IL-7 (pg/mL)	462.4 ± 53.32	380.9 ± 66.48	208.9 ± 64.7	261.4 ± 70.09	*p* = 0.008	ns	ns
IL-10 (pg/mL)	22.49 ± 4.88	22.45 ± 5.84	11.97 ± 1.11	14.93 ± 1.62	*p* = 0.03	ns	ns
IFN-γ (pg/mL)	39.6 ± 4.76	31.1 ± 6.76	28.5 ± 4.96	39.79 ± 5.46	ns	ns	ns
RANTES (pg/mL)	28.87 ± 2.11	24.5 ± 1.46	24.9 ± 1.85	33.3 ± 3.32	ns	ns	*p* = 0.008
TNF-α (pg/mL)	374.5 ± 59.84	254.8 ± 47.68	286.5 ± 12.68	327 ± 20.54	ns	ns	ns
VEGF (pg/mL)	10.55 ± 1.58	9.13 ± 2.70	9.2 ± 2.09	11.03 ± 2.16	ns	ns	ns
MCP-1 (pg/mL)	91.65 ± 8.63	77.86 ± 12.44	93.04 ± 5.53	94.36 ± 10.44	ns	ns	ns

LLA: low linoleic acid; HLA: high linoleic acid; IL: interleukin; IFN-γ: interferon-gamma; RANTES: regulated on activation normal T cell expressed and secreted; TNF-α: tumor necrosis factor- alpha; VEGF: vascular endothelial growth factor; MCP-1: Monocyte chemoattractant protein-1. Placentae from one male and one female per litter were analyzed. Data expressed as mean ± SEM and analyzed by two-way ANOVA with diet and sex as two factors. *n* = 5–8.

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
