# Peer review of "Maternal High Linoleic Acid Alters Placental Fatty Acid Composition"

_nutrients, 2020, doi:10.3390/nu12082183_

Round 1
Reviewer 1 Report
The manuscript by Shrestha and coll. reports the effect of increased maternal linoleic acid on placental fatty acid storage, metabolism and transport, and general placental function.
Just a few minor considerations:
In the abstract, what is the extended name of the acronyms Angptl4 and Daglb?
The authors have to better analyze the text, as in the case of the sentence in the discussion: "Of note, the pattern of change in placental fatty acid composition in the placenta was similar to both the maternal and offspring circulations, with some exceptions "in which placental is repeated.
Author Response
Reviewer 1
The manuscript by Shrestha and coll. reports the effect of increased maternal linoleic acid on placental fatty acid storage, metabolism and transport, and general placental function.
Just a few minor considerations:
|
SN |
Comments |
Responses |
|
1. |
In the abstract, what is the extended name of the acronyms Angptl4 and Daglb? |
Thank you for the suggestion. Extended names have been included in Line 27-28. |
|
2. |
The authors have to better analyze the text, as in the case of the sentence in the discussion: "Of note, the pattern of change in placental fatty acid composition in the placenta was similar to both the maternal and offspring circulations, with some exceptions "in which placental is repeated. |
The repeated word is removed in Line 333. This has been checked throughout the manuscript. |
Reviewer 2 Report
The study is well described, and the results are interesting, adding information regarding the potential impact of a diet high in LA (6% of energy as in the typical Western diet) on fatty acid storage, metabolism and transport related to an important period in development. The discussion could benefit from a better distinction between human and animal research. Also, the authors should clarify in the discussion how their findings relate to current knowledge of fatty acid intake and changes in the human placenta.
Minor comments:
Abstract, L 27. Abbreviations Angptl4 and Daglb
Introduction:
L 41. I find the statement “oxidization of eicosanoids is associated with disease development” misleading. Eicosanoids are mediators generated by oxidation of AA, some of which are important in physiological processes, whereas some are associated with disease development.
It would improve the understanding of the study if you were to include a figure of the metabolism and degradation of LA and ALA in the introduction.
Methods:
L 86. The details of the intervention are very important, and the diets need to be better described in this manuscript. Referral to a previous study is not sufficient. For example, the percentages given in the abstract should also be stated in the methods.
Results:
The results are clearly presented and the tables easy to follow. The groups are difficult to distinguish in Figure 1 A-B.
270-272 This sentence does not belong in the results section, is not clear, and needs a reference.
272-275 Clarify that the results refer to cytokines in the placental labyrinth zone.
Discussion:
The discussion is generally well balanced regarding all different findings but could be improved as stated above with added information from recent human studies.
L 290-293 There were clear differences in placental fatty acid composition, but related to transport and angiogenesis only a few differences in gene expression were shown and this should be reflected in the initial statement in the discussion.
L 294 The outcome in this study was indirectly associated with the offspring but the results demonstrated were not really offspring outcomes.
L 327 I would add to the discussion also the known importance of Fads2. (for example Steer et al. 2012 Polyunsaturated Fatty Acid Levels in Blood During Pregnancy, at Birth and at 7 Years: Their Associations With Two Common FADS2 Polymorphisms; Lattka et al. 2013 Umbilical cord PUFA are determined by maternal and child fatty acid desaturase (FADS) genetic variants in the Avon Longitudinal Study of Parents and Children (ALSPAC))
L 348 HFD abbreviation not previously used
L 365-368 Example of a section where results from human and animal studies are stated without making the context clear to the reader.
L 389-403 This section is a bit hard to follow and could be improved.
L 407 “high maternal diet”, missing LA
Author Response
Reviewer 2
The study is well described, and the results are interesting, adding information regarding the potential impact of a diet high in LA (6% of energy as in the typical Western diet) on fatty acid storage, metabolism and transport related to an important period in development. The discussion could benefit from a better distinction between human and animal research. Also, the authors should clarify in the discussion how their findings relate to current knowledge of fatty acid intake and changes in the human placenta.
Minor comments:
|
SN |
Comments |
Responses |
|
1. |
Abstract, L 27. Abbreviations Angptl4 and Daglb |
Extended names have been included in Line 27-28. |
|
2. |
Introduction: L 41. I find the statement “oxidization of eicosanoids is associated with disease development” misleading. Eicosanoids are mediators generated by oxidation of AA, some of which are important in physiological processes, whereas some are associated with disease development. |
Thank you for the comment. The sentence has been rephrased in Line 47-49 as suggested. |
|
3. |
It would improve the understanding of the study if you were to include a figure of the metabolism and degradation of LA and ALA in the introduction. |
For the understanding, figure showing LA and ALA metabolism is attached (Line 61). |
|
4. |
Methods: L 86. The details of the intervention are very important, and the diets need to be better described in this manuscript. Referral to a previous study is not sufficient. For example, the percentages given in the abstract should also be stated in the methods. |
The information about the diet (% of energy from LA) has been included in method section (Line 105-106). |
|
5. |
Results: The results are clearly presented and the tables easy to follow. The groups are difficult to distinguish in Figure 1 A-B. |
Thank you for the suggestion. The figure has been colored to distinguish the different groups (maternal diet X-axis, postnatal diet with symbol of dot plot). Now, Fig 1 is Fig 2. |
|
6. |
270-272 This sentence does not belong in the results section, is not clear, and needs a reference. |
The references have been added to make it clear in Line 295 and 297. |
|
7. |
272-275 Clarify that the results refer to cytokines in the placental labyrinth zone. |
This has been included in line 300 for clarification. |
|
8. |
Discussion: The discussion is generally well balanced regarding all different findings but could be improved as stated above with added information from recent human studies. |
Animal model or human study is specified where needed. Human studies are included when possible (Line 361-362). Recent human study is also included (line 359-360, Nita R et al 2020). |
|
9. |
L 290-293 There were clear differences in placental fatty acid composition, but related to transport and angiogenesis only a few differences in gene expression were shown and this should be reflected in the initial statement in the discussion. |
The suggested statement is included in Line 318-321. |
|
10. |
L 294 The outcome in this study was indirectly associated with the offspring but the results demonstrated were not really offspring outcomes. |
This has been corrected as suggested in line 322-323. |
|
11. |
L 327 I would add to the discussion also the known importance of Fads2. (for example Steer et al. 2012 Polyunsaturated Fatty Acid Levels in Blood During Pregnancy, at Birth and at 7 Years: Their Associations With Two Common FADS2 Polymorphisms; Lattka et al. 2013 Umbilical cord PUFA are determined by maternal and child fatty acid desaturase (FADS) genetic variants in the Avon Longitudinal Study of Parents and Children (ALSPAC)) |
The association between PUFA and Fads2 has been discussed in discussion section as suggested (Line 362-364). |
|
12. |
L 348 HFD abbreviation not previously used |
HFD is now defined in line 382. |
|
13. |
L 365-368 Example of a section where results from human and animal studies are stated without making the context clear to the reader. |
Human and animal model have been mentioned in the statement for clarification in line 400-405. |
|
14. |
L 389-403 This section is a bit hard to follow and could be improved. |
The sentences in the paragraph have been rephrased for the improvement (line 436-441). |
|
15. |
L 407 “high maternal diet”, missing LA |
The missing word has been added (Line 447). |